# Time-on-task estimation for tasks lasting hours spread over multiple days

**Kaden Hart**[1]*, **Christopher M. Warren**[2], **John Edwards**[1]

**1** Department of Computer Science, Utah State University, Logan, Utah, United States of America,
**2** Department of Psychology, Utah State University, Logan, Utah, United States of America

* kaden.hart@usu.edu

**Data availability statement:** Data cannot be shared publicly because it is identifiable. De-identification of programming process data is difficult and anonymization is nearly impossible to prove. This is because the

## Abstract

The ability of a human to retrospectively estimate the amount of time spent on a task is largely only understood when the period of time is seconds- or minutes-long. The lack of research into estimation of longer periods of time can be attributed, in part, to the difficulty of measuring ground truth durations when the task is broken up by other activities in a natural, day-to-day setting. An empirically based model of engagement was recently proposed that statistically estimates time-on-task for computer programming assignments in an introductory computer programming course. Computer programming assignments can be completed in many sessions across days or weeks and, based on recorded keystroke data, an objective ground truth of task duration can be measured. In this work, we take advantage of this new measurement method to explore duration estimation of tasks lasting hours that are spread out over multiple days in a natural setting. Subjects in our study overestimated time-on-task 78% of the time and reported a median of 1.45 hours worked for every actual hour spent on task. We find that self-reports are more accurate when students score higher on their assignments in our data, suggesting the accuracy of estimated time is correlated with task performance.

## Introduction

Time duration estimation, where a person estimates how much time elapsed during an interval or while completing a task, is much-studied because of its impacts on daily life and on our psychological and physiological understanding of related phenomena, such as memory. Time estimation studies generally take the form of comparing a subject's estimate of time duration with an established actual time duration, measuring how much the subject over- or underestimates duration.

There are two broad classes of time duration studies, those in which duration estimates range from seconds to minutes, and those in which estimates are in hours. Studies that use durations at the second and minute levels are generally called interval timing studies [1,2]. These studies merge the theory of time awareness with empirical studies and with physiology. They are generally conducted in highly controlled laboratory conditions and often seek to isolate specific mechanisms, so few (usually one) tasks or stimuli are allowed to occur in the

keystrokes logs contain a record of every edit a student made to a document, which includes their name. Students could add any type of identifiable information to their homework and later remove them, but the log will still contain that information. We can remove this but if they have typed their name or other identifying information elsewhere. Even careful inspection can miss potentially identifiable information. USU's institutional review board can be contacted at irb@usu.edu or 435-797-1821.

**Funding:** The author(s) received no specific funding for this work.

**Competing interests:** The authors have declared that no competing interests exist.

estimated time interval. Most studies use durations no longer than a few seconds, but some extend to as long as 60 minutes. Ground truth time duration is simply clock time.

The other broad class of time estimation study is of interest to the labor and organizational psychology communities. These studies are often conducted in the context of understanding employee effort and performance and treat much longer time durations, with subjects estimating time spent on a task over the course of a day or a work week. Ground truth time-on-task is difficult to measure, so most studies use time diaries, where subjects recount all tasks done during the day, as a subjective stand-in for actual time spent on a given task.

A gap exists between these two methodologies. On the one hand, we have highly controlled studies with trustworthy ground truth, but they study only relatively short time intervals and they are generally conducted in laboratory settings. On the other hand, we have loosely controlled studies with subjective ground truth with little theoretical or physiological explanation, but they span longer time intervals and are conducted in natural settings.

In this paper, we report a study that is a step in bridging this gap. We extend task lengths to durations of hours and allow task switching. Our data is collected in a natural (not laboratory) setting while using an objective measure of actual time duration that we can compare subjects' estimates to. We build on work recently published in the computing education literature that proposes a way to accurately measure how long computer programming students work on an hours-long assignment using keystroke logs [3,4]. These time estimates work by recording students' keystrokes while they work on computer programming assignments whenever and wherever they choose, then use a probabilistic model to determine how much time students spend working on their assignments. With these keystroke-derived time-on-task measurements we can compare students' retrospective estimates of time-on-task for a given assignment. Using this methodology we gain benefits from both classes of duration estimation studies: we use objective ground truth measurements like interval (short duration) timing studies and we are able to measure in natural settings for longer periods of time, like in time diary studies. Furthermore, the task need not be completed in one continuous session. We incorporate an additional variable, that of performance as measured by graded scores on assignments, in our analysis.

Our overarching research question is: how accurately do humans estimate time-on-task after completing the task, and how does this correlate with performance? Our contribution in this paper is to take the first measurements in a natural setting where the baseline time estimates do not rely on self-reports, but are objective estimates based on a statistical model of behavior. Our specific research questions in this study are:

RQ1 *How accurately do students estimate time spent on computer programming assignments?*

RQ2 *Does a correlation exist between performance and the accuracy of self-reported task duration?*

We find that subjects overestimate their assignment duration by a median of 45%, reporting 1.45 hours for every hour worked; confirming the feeling of CS1 instructors that student estimates felt inflated [3]. The 45% error is much higher than what was found in time diaries. Despite the large overestimates, the majority of student participants nevertheless believed themselves able to accurately estimate time-on-task according to a survey. We also find that performance on assignments is moderately correlated with the accuracy of time-on-task estimates. This study is the first to compare estimated time-on-task to a baseline obtained using an objective, empirically based method that is measured in a natural non-laboratory setting.

The experiment is done in an educational setting, but we suggest that the results are potentially generalizable and important enough to contribute to the general time-on-task estimation literature. This paper contributes to time-estimate research by demonstrating methodology to obtain objective measures for long duration tasks in a case study exploring subjective time estimates to objective time estimates.

## Background

### Interval time estimation

Interval timing is tracking elapsed time in the seconds to minutes range [1]. It is typically done in a laboratory setting where actual time duration is simply clock time since the intervals are short enough that subjects don't have time to task switch. There are two paradigms of duration estimation, prospective and retrospective. In prospective estimation, participants are told beforehand that they will be asked about time, while retrospective estimates are made without prior knowledge that a time estimate will be made [5]. When participants know they will be asked about time they tend to overestimate more [6], although Walker et al. [7] only found this effect at 8 minutes and not at longer durations. Balci et al. [1] found in a retrospective study with approximately 24,500 participants and intervals from 5 minutes to 90 minutes that intervals less than 15 minutes tended to be overestimated and intervals over 15 minutes tended to be underestimated.

Tobin et al. [6] found that gamers overestimate their time spent gaming at all intervals tested: 12 minutes, 35 minutes, and 58 minutes. It is well known that "time flies when you're having fun" [8] but gamers did not underestimate their time; possibly because they are aware that their perceived time is less than their actual time [9] and increase their estimates to compensate. If the task chosen for the interval was not as engaging (e.g. homework), overestimation could be higher because the perception of time is inflated when tasks are not engaging [10].

An important feature of interval timing is the existence of the objective ground truth interval duration to compare to estimates, yet there is a "paucity" [7] of prospective vs. retrospective interval timing studies that consider intervals longer than two minutes [6].

### Time diary studies

Related to interval timing are studies that measure time on tasks lasting hours spread over multiple days. These studies are important to labor researchers and stakeholders; however an accurate ground truth is rarely available. Time estimation studies where durations are in the range of hours use various measurement and estimation methods. The most common is the time-diary study. In these studies, participants keep a daily "time diary," where they log each activity done sequentially during the day. This is used as a stand-in for actual time duration for a specific task. The estimated time comes after a number of days (usually a week) and is compared to the time diaries. Time estimates generally exceed estimates derived from time diaries. Other measurement methods at this time scale exist, but time diaries are generally accepted as a "richer and more contextualized source of information about people's activities than any present alternative" [11].

Time-diaries provided by various universities starting with the University of Michigan in 1965 and since 2003 by the US Census Bureau with the Bureau of Labor Statistics American Time Use Survey have been used as ground truth [11]. Time diaries have their issues: people can distort, embellish, lie, forget, or "substitute a habitual activity for what actually took place" [11]. Despite their issues, time-diaries are used because they are widely available,

have a long history, encompass almost every profession, and are the most accurate measures available for their broad contexts [12–14].

It is well documented that people overestimate their time spent working when comparing time estimates to time-diaries [15–17] by 5% to 10% [11] especially for mundane tasks like housework [18,19], although there exists some debate on the interpretation of these overestimates (e.g. are respondents including housework or commute time when asked about work in surveys?) [20].

## Measuring time-on-task using keystroke logs

In an educational context, the amount of time spent working on a task has been researched since the 1970's [21]. It is an important metric that contributes to learning and achievement [22]. To measure time-on-task, computing education researchers often use software that records events when students work on their computer code. Course-grained measures, such as measuring time-on-task as the time between when a student started working on their assignment and when they finished working on their assignment, are not as accurate as fine-grained measures, such as using keystrokes to determine when a student was working or not [22]. The time between two keystrokes is known as the keystroke latency.

A straightforward way to estimate time-on-task using keystroke logs is to simply sum the keystroke latencies. However, if a latency of, say, two hours appears, then we intuitively consider that the student was taking a break at that time and don't include that latency duration in the time-on-task estimate. Threshold-based time estimation methods define a threshold latency value such that any latency greater than the threshold is excluded from the time-on-task estimate. For years, ad hoc thresholds were used, which ranged from five minutes to one hour [23–27]. This lack of consistency in choice of threshold has been a major issue [3,4,21]. Without a validated way of determining which latencies represented disengagement from the task, time estimates from event logs could not be trusted.

Very recently, Edwards et al. [3] and Hart et al. [4] developed a statistical model that could probabilistically determine how likely a latency is to indicate the student was engaged or not (e.g., short latency keystrokes were more likely to indicate engagement than long latency keystrokes). Using this model, an accurate estimate could be made for how much time a student worked on an assignment by summing the keystroke latencies that are determined to be engaged by the model.

During the study developing the statistical model of engagement, Edwards et. al. noted that when students are asked how long an assignment took to complete, student responses seemed inflated to the instructor [3]. This work investigates the validity of that notion.

## Bridging the gap

With our methodology, we start to explore the space between interval timing and time diary studies. For example, in one interval timing study using a retrospective estimation paradigm, subjects tended to overestimate when the time interval is less than 15 minutes and were found to underestimate when the interval was between 15 and 90 minutes [1]. This is in contrast to a time diary study in which time diaries (the surrogate for ground truth) showed more working hours when subjects estimated fewer than 25 hours worked in a week but showed fewer working hours when weekly estimates exceeded 25 hours [11]. A direct comparison between these two studies is not possible because of the major methodological and contextual differences.

We view our time-on-task estimation using the lens of retrospective timing processes. Technically, the experiment is one of prospective timing, since for the majority of assignments, participants know that they will be asked to estimate afterwards how long they took to

complete a programming assignment. However, because of the scale and length of task, and because the task is not purposed to the time estimation study, but rather, estimating time-on-task is more of an after-the-fact add-on, we suggest that the processes involved in time estimation are more closely aligned with retrospective timing. This is supported by our results: if we make the assumption that lower-performing students, in terms of assignment score, experience higher cognitive loads, then, according to [28], the duration judgment ratio (subjective duration to objective duration) should increase in a retrospective setting. In other words, if subjects are not informed ahead of time that they will be estimating the time taken, they tend to overestimate the time when under cognitive load. However, this could also be explained by the effect of prospective/retrospective estimation paradigm becoming less pronounced as length of task increases [7].

## Methods

Thirty-three students in an introductory computer programming course (CS1) at Utah State University, a mid-sized, research-intensive university in the western United States, participated in our study. The CS1 course is a semester-long (14 weeks) course that teaches students the fundamentals of writing computer programs in the Python programming language. As a general science elective, the course attracts a mix of computer science majors and non-majors as well as a mix of students with prior programming experience and those without. Students were given a short, 2-minute, description of the study then given an informed consent document to sign and return to the research team if they decided to participate. Students were recruited from 08/05/2023 to 13/10/2023. Data were collected during the Summer (May 8th to August 18th) and Fall (August 28th to December 15th) semester of 2023. Our university's ethics review board reviewed and approved this study (IRB #13514).

In the CS1 course, students must complete approximately one programming assignment each week. In this study, we report data on seven of the assignments. Four of the assignments have two parts, or "tasks." See Table 1 for brief descriptions of assignments, each of which targets a specific concept or concepts in computer programming (e.g., iteration, conditionals, etc). We collected data on 106 assignment submissions, but removed one submission because there was not enough keystroke data collected during the assignment; we used the remaining 105 submissions for all analyses. Students write their code using a specialized piece of software called an integrated development environment (IDE). IDEs provide a convenient environment that includes text coloring, debugging tools, code suggestions, and a user interface organized around the file structure of the code files. Students in our study were encouraged to use the *PyCharm* IDE.

### Measurement of actual time duration

The primary data used in this study is keystroke data. Study participants were asked to install a plugin to their *PyCharm* IDE called *ShowYourWork* [29,30]. This plugin logs keystrokes, pastes, and other events to a file. Each entry in the file contains a timestamp, the action type (e.g. keystroke), any text that was inserted or deleted as a result of the action, and other data. This file would be updated anytime a student worked on their assignment and included in their final submission.

The key to this study is our ability to objectively measure how long students took to complete their programming assignments without requiring them to work in a laboratory setting. Students could work on their assignments on campus or at home at any time that they chose. They could take breaks and even work on the assignment across multiple days (most did,

**Table 1. Descriptions of programs students programmed for their assignments.**

| Assignment | Task | Description |
|---|---|---|
| A1 | 1 | Type a year into the command prompt and the program reports if the year is a leap year or not |
| A1 | 2 | The program prompts the user for information about animals seen then generates a summary with indenting for legibility. |
| A2 | 1 | The program asks simple addition problems for the user to solve and animates a score for answers with more points awarded or lost based on how quickly the answer was given. |
| A2 | 2 | The program prints a properly formatted number pyramid when given the number of rows. |
| A3 | 1 | Draw an 8x8 black and white grid to the users specified width and height. |
| A3 | 2 | Draw random numbers of rectangles or circles evenly spaced and rotated in circular patterns at random positions. |
| A4 | 1 | A game where the player must move around to collect treasure and avoid a moving enemy. |
| A4 | 2 | Draw a yellow smiley face that can be changed to a frown or a different color. |
| A5 | 1 | A Duck Hunt like game where the player must catch butterflies and kill wasps. |
| A6 | 1 | A Snake like game with two players. A player loses when they run into a wall or the path left by themselves or the other player. |
| A7 | 1 | Given a program that makes a deck of cards and deals them correctly, find and fix errors in the given sorting and searching code. |

in fact). Our time-on-task measurement uses keystroke log data obtained using the *ShowYour-Work* plugin. As discussed in the Background section, recent work in the computing education literature has established an empirically based statistical model that gives probabilities that students are on task at any given moment based on the amount of time elapsed since their last keystroke [3,4]. Using these probabilities, we can estimate, in the aggregate, the amount of time spent on the assignment. This is done with the student in a natural setting. Furthermore, the Hawthorne effect [31] is minimized by the fact that roughly half of the students forgot that their keystrokes were being logged [30].

To estimate how much time a student spent working on an assignment, we used the equation from the work of Hart et al. [4]

$$y(x) = \begin{cases} 1 & \text{if } x < 0.75 \\ \frac{1}{(1+Qe^{-B(x-M)})^{1/v}} & \text{if } x \in [0.75, 120] \\ 0 & \text{if } x > 120 \end{cases} \tag{1}$$

with $Q = 6604$, $B = -4.99$, $M = 0.01$, $v = 58.32$, and x equal to the keystroke latency in minutes. This equation gives the probability that a student was working on their assignment during the time between two keystrokes. With this equation, each submission was probabilistically sampled many times to create a range of possible assignment durations. The median sampled assignment duration was then used as the true assignment duration.

## Time estimation

The course had a post-assignment reflection questionnaire for each assignment that asked students questions about the assignment for instructor feedback but also included a question for our research: "*How many hours did you spend working on this assignment? (example: 2.5, 1.25, 4)*". Students were required to enter an estimate to receive credit on the questionnaire.

### Duration judgment ratio

Our main analysis measure is the so-called duration judgment ratio, or simply ratio, which is a measure of quality of duration estimation [28]. The duration judgment ratio is defined as $E_D/T_D$ where $E_D$ is the estimated duration and $T_D$ is the actual task duration. The range of the ratio is $[0, \infty)$. Estimated durations come from the post-assignment surveys and actual task durations are derived from the keystroke data.

### Pre-survey

Additionally, participants were given a survey at the beginning of the course:

- I am capable of remembering how much time I worked on an assignment. [*Strongly Disagree, Slightly Disagree, Neither Agree nor Disagree, Slightly Agree, Strongly Agree*]
- A 3-hour assignment is: [*Significantly Shorter Than Most, Slightly Shorter Than Most, About Average, Slightly Longer Than Most, Significantly Longer Than Most*]

## Results and discussion

### Estimated time

Our first research question RQ1 is: *How accurately do students estimate time spent on computer programming assignments?* In our data, students over-reported their assignment durations 78% of the time with an average ratio of 2.2, and a median ratio of 1.45. This means that students reported an average of 2.2 hours worked for every hour actually worked. As seen in Fig 1, outliers, such as the student who reported 8 hours for each hour actually worked, skewed the distribution. The median of measured assignment time-on-task in our data is 5.1

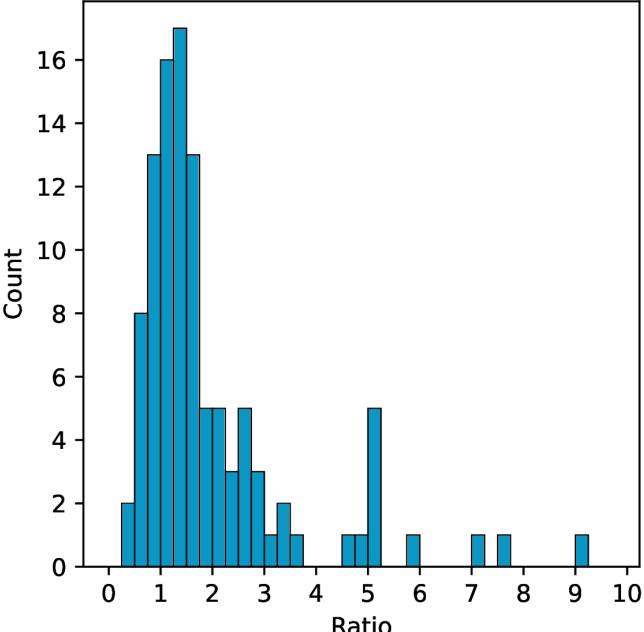

**Fig 1. Distribution of duration judgment ratios between estimated duration and actual duration for each assignment submission.** Most estimates resulted in a ratio greater than one, meaning the student over estimated how long they took on the assignment.

hours, with a median error of 3.2 hours, i.e., the median reported assignment time was 8.3 hours for a 5.1-hour assignment.

The size of our subjects' estimation error is significantly larger than the 5% to 10% found by Robinson et al. [11] in a time diary study. This may be because students can work on assignments with extreme flexibility and greater errors and exaggerations have been found when people have nonstandard or irregular schedules [32]. Homework may also be more stressful to students because of mismanaged time, difficult deadlines, and inexperience leading to higher error. Estimates in industry may be more accurate because there may be a known expected answer, such as a typical 40-hour workweek, where an assignment has no expected duration. It is also likely that students did not account for their wasted time due to short-duration breaks such as phone notifications. Time diary study ground-truth durations likely include short breaks [15], while our actual duration measures do not count breaks. This leads to what we suggest as the most likely cause of difference in estimate error between our study and time diary studies: we use an objective measure of actual time duration while time diaries rely on self-report. The implication is that the time estimation error in time diary studies may be greatly underestimated.

## Task performance vs. Estimate error

Our second research question RQ2 is: *Does a correlation exist between performance and the accuracy of self-reported task duration?* We analyzed two measures of performance. The first measure is assignment score. The time duration ratio had a moderate negative correlation with assignment score (Spearman's rank correlation $r = -0.4$, $p < 0.001$ Table 2). See Fig 2. In other words, when students received a higher grade, their self-reported duration was more accurate. The average score for an assignment where a student underestimated their time was 94% and the average score for overestimated submissions was 81%. As a second measure of performance, we considered total time-on-task and we correlated the total amount of time a student took to complete an assignment with time duration ratio. We found a similar moderate negative correlation (Spearman's rank correlation $r = -0.46$, $p = 1.8e^{-6}$ Table 2). See Fig 3. If we assume that students getting lower grades or those taking longer experience more cognitive load, then these findings are consistent with previous work that shows that perceived duration is dependent on cognitive load [28,33]. Difficulty has also been shown to disrupt time estimation [34] and frustration also inflates time estimates [10].

As a related measure, we looked at a possible correlation between "break ratio" and time duration ratio. The break ratio is the percentage of keystroke latencies that are greater than 60 seconds. Based on work done in interval timing that suggests that retrospective duration judgments lengthen with more remembered context changes [33], we hypothesized that the correlation would be positive, since a higher break ratio implies more breaks and thus more context changes. While the hypothesis test suggests a possible weak correlation, the finding is not statistically significant (Spearman's rank correlation $r = 0.15$, $p = 0.13$ Table 2). See Fig 4.

**Table 2. Correlation tests.**

| Variable | Variable | Test | Coefficient | p-Value |
|---|---|---|---|---|
| Error Ratio | Assignment Score | Spearman | –0.4 | <0.001 |
| Error Ratio | Time-On-Task | Spearman | –0.46 | <0.001 |
| Error Ratio | Break Ratio | Spearman | 0.15 | =0.13 |
| Error Ratio | Estimation Confidence | Kendall | 0.01 | =0.93 |
| Error Ratio | 3-Hour Perception | Kendall | 0.03 | =0.85 |

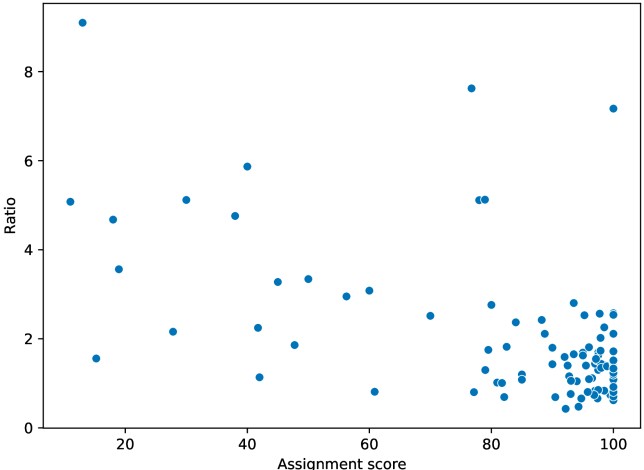

**Fig 2. Assignment score vs. Duration judgment ratio.**

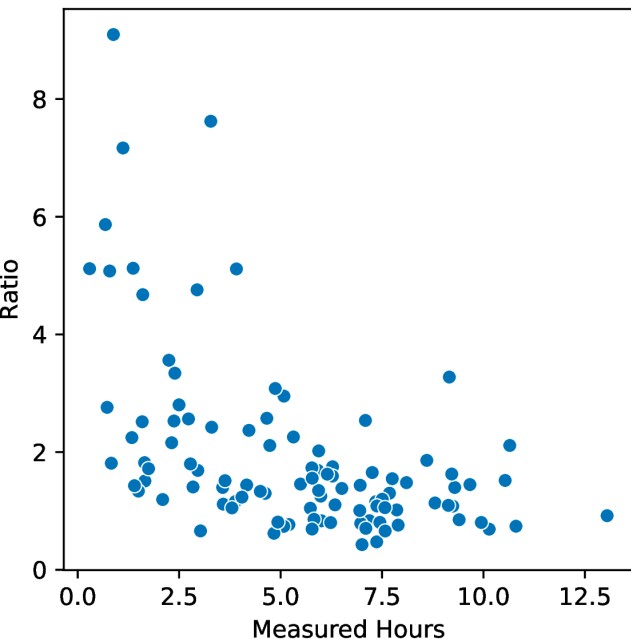

**Fig 3. Total measured time vs. Ratio.** Since some of the students with very small measured times may have worked outside of the *PyCharm* IDE that we used to capture keystrokes, we performed the test throwing out submissions that measured less than one hour of working time.

## Analysis of error in actual durations

An issue with estimating total assignment duration from keystroke data is that individuals may act very differently from each other, but we only use one model for engagement for all students [4]. While the model is robust for a group of students, it is not validated at an individual level. To account for error in individual differences we made two confidence intervals for each assignment submission. The first interval we built was the 100% range, meaning the

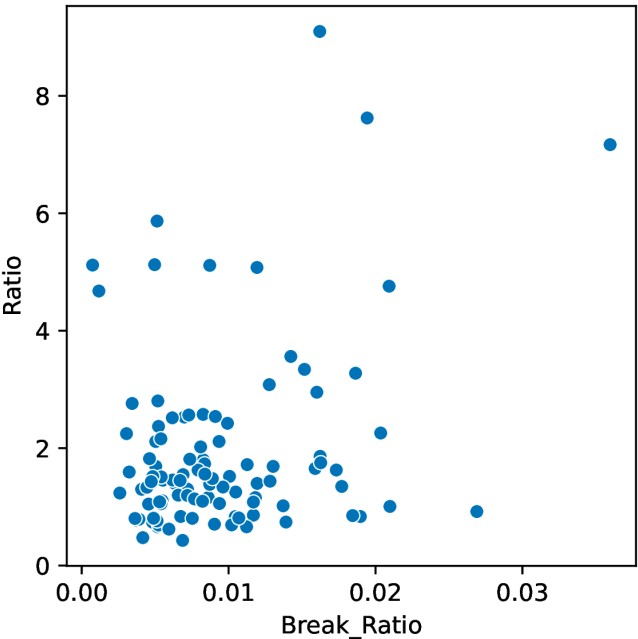

**Fig 4. Breaks vs. Ratio.**

absolute minimal and maximal duration supported by the data. The minimum time an assignment could take, based on Hart et al.'s [4] model, was the sum of the keystrokes with a latency of 45 seconds or less. Similarly, the maximum time an assignment could take was the sum of the keystrokes with a latency of 2 hours or less. This yielded an average interval of 6.67 hours. Surprisingly, we found that about half (49%) of self-reported assignment durations fell outside this 100% confidence interval, giving strong supporting evidence that students are, in fact, overestimating time-on-task.

The second confidence interval we made was the 90% interval. We chose this range because it is still broad enough to allow for individual differences in behavior, but doesn't include the extremely unlikely cases that the 100% interval does. It is impractical to find the true 90% confidence interval because there are on average 115 latencies that must be probabilistically defined as engaged or disengaged, and thus $2^{115}$ possible durations for each assignment. To estimate the 90% confidence interval we probabilistically sampled each assignment's duration 1,000 times, then found the 90% interval of our 1,000 samples. We chose a sample size of 1,000 by testing varying sample sizes from 100 to 2,000 and investigating the variance between samples. A sample size of 1000 provides reasonably small variance and could be computed quickly. See Fig 5. We found an average range of 1.05 hours for the 90% confidence interval and considered an estimate accurate if it falls within this interval. With this smaller range of uncertainty, we found only 9.5% of assignment submissions with reported times within the 90% confidence interval. See Fig 6. In our data, we find that self-reported assignment durations are not only inaccurate but inflated beyond reason with only 9.5% considered accurate.

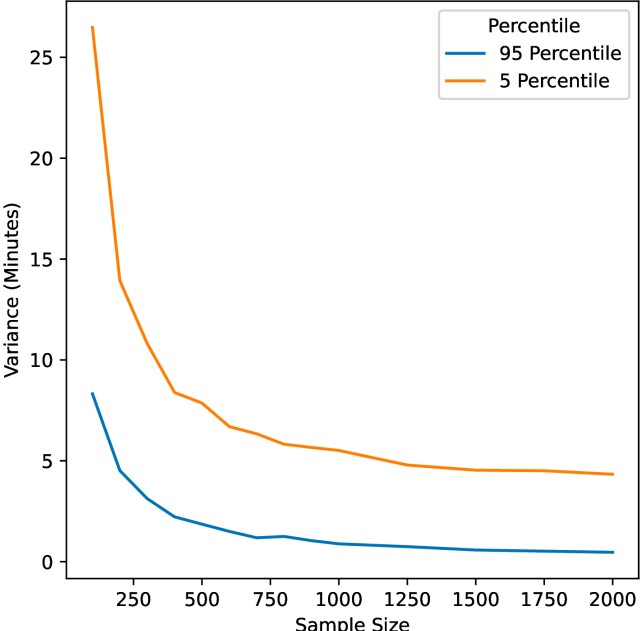

**Fig 5. Variance vs. Sample size for probabilistically sampled assignment durations.** We ran 50 iterations of each sample size and computed the variance across iterations. For example, in sample size 1000 we estimated each assignment duration 1000 times, then calculated the percentiles. We repeated this 50 times and computed the average variance in all assignments.

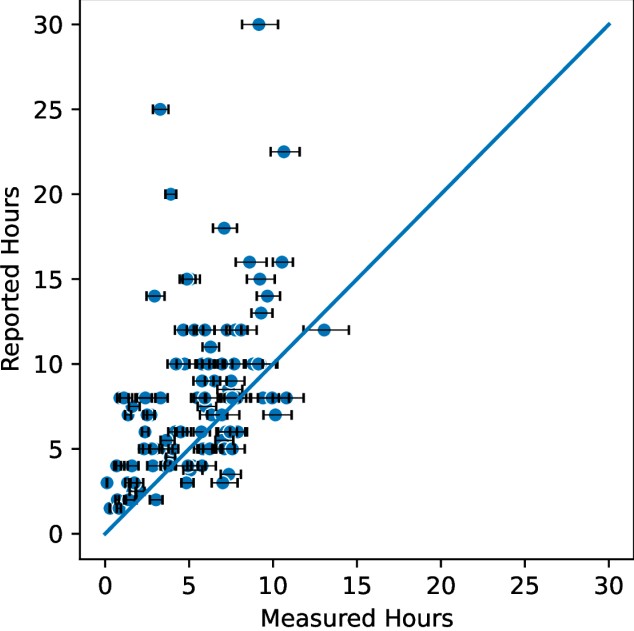

**Fig 6. Measured vs. Reported assignment duration with 90% confidence interval for assignment submission.** The center line indicates where estimated duration equals actual duration.

### Pre-survey

In response to the question "I am capable of remembering how much time I worked on an assignment," 79% of students selected either "Agree" or "Strongly Agree" in their ability to track time (Fig 7). Students think they can keep track of time spent working, but rarely can. This has implications that the memory of time spent working on assignments, and possibly the recollected effort and lost free time, is greater than reality. The perception of doing homework could be worse than the reality of doing homework. Understanding this could help motivate students to get their work done because it's not as bad as they think it is. Students' confidence in their ability to track time was not correlated with the estimation ratio (Kendall rank = 0.01, $p$ = 0.93 Table 2).

In response to the question asking about the duration of a 3-hour assignment, 48% of students indicated that a 3-hour assignment was longer or significantly longer than most, and 48% indicated that a 3-hour assignment was about average (Fig 8). The median assignment time in our data was 5.1 hours; 70% longer than students were surveyed about. It is likely that students knew their assignments were lasting longer than what they thought was normal for most assignments causing some overestimation because assignments felt longer than most. Students' perception of the duration of a 3-hour assignment was not correlated to the estimation ratio (Kendall rank = 0.03, $p$ = 0.85 Table 2). Future work could investigate higher level courses: sophomore, junior and senior. Estimates may improve because students are accustomed to the college environment and have learned better time management skills. Overestimating assignment durations may be a good thing for students because they will plan more time to get their work done, but it is also possible that they do not plan enough time due to the planning fallacy [35].

### Threats to validity

A possible threat to validity is that the model of engagement does not account for work done before a student's first keystroke. Additionally, students make their time duration estimate in a post-assignment survey which could be done in a setting that could affect their estimate [36].

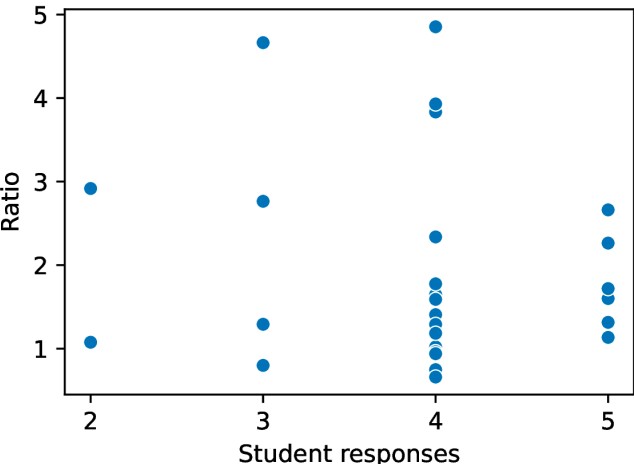

**Fig 7. Average ratio and survey responses for the survey question "I am capable of remembering how much time I worked on an assignment."** Where 1 = Strongly Disagree, ..., 5 = Strongly Agree.

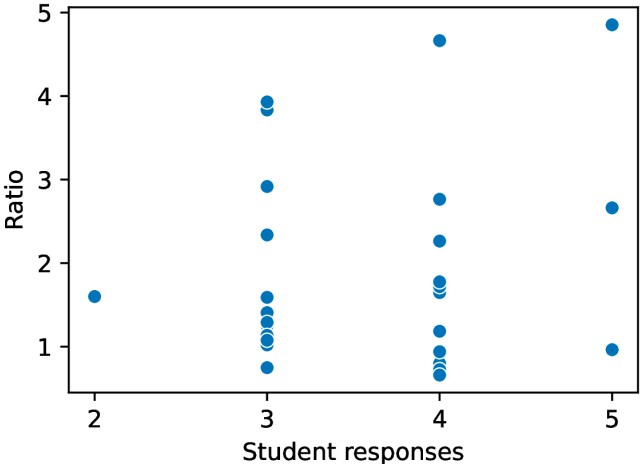

**Fig 8. Average estimation ratio and survey responses for the survey question "A 3-hour assignment is: 1 = Significantly Shorter Than Most, ..., 5 = Significantly Longer Than Most."**

A shortcoming of any self-reported data is "that self-report and objective measures provide information on distinct, different aspects of work performance" [14] (e.g., counting the commute to work as work time). In our study, students may count the time spent downloading starter files, setting up their IDE, and submitting an assignment where we consider such tasks to be unrelated to the learning objectives of the assignment and therefore not measured as time-on-task by our model. (A member of the research team timed themselves doing these steps at under 3 minutes.) This discrepancy could explain some of the error between self-reported and measured assignment durations.

Students also exhibited a rounding bias, rounding estimates to whole or half hour increments; our measured time-on-task has millisecond resolution. We controlled for this with analysis of confidence intervals and giving students a generous range to be considered accurate.

This work was done in a single University's CS1 course. The results of this study may not hold in other contexts.

## Conclusions

This paper reports results of a study in which CS1 students estimated how much time they spent working on their assignments. Students overestimate time-on-task in CS1 programming tasks even across multiple assignments and contexts in which students make the duration estimation, all of which should factor into the complex interactions affecting duration judgments [36]. We found that students think they can accurately remember how much time they spend working on their assignments, yet students overestimated time-on-task 79% of the time, usually by 45% or more, which is much higher than overestimates found in working time studies. We also find that when a student scores higher on an assignment or takes less time to complete the assignment, their self-report is more likely to be accurate.

This work is a first step in bridging the gap between second- and minute-level interval time estimation studies and hour- and day-level time diary studies. Our ability to objectively measure ground truth durations while subjects perform a task in a natural setting at times of their choosing further emphasizes the validity of our results.

Our study also enables research into whether additional theories of time awareness apply at larger time scales. For example, it has been suggested that the ability of a human to make retrospective duration judgments isn't based just on how well they are able to recall individual events, but rather, they use a heuristic based on how easily they can retrieve the events of the time period [37]. Our work takes the next step in validating the heuristic hypothesis and describing the heuristic itself. Some work suggests that time estimate error may behave similarly to the senses such as vision and hearing [38,39]. Those studies were conducted in the second and minute range. Using our high temporal resolution ground truthing could uncover whether this behavior scales to tasks lasting hours.

Estimates are influenced by expected completion time [11]. For future work, a study varying expected completion time would be important. This could be done by splitting student participants into three conditions and telling one group that the assignment would be expected to be completed in one time duration, telling another group another duration, and not giving the third group any time estimate. In other future work, performing similar studies in other contexts is needed. This would require a context in which high temporal resolution events (e.g., keystrokes) can be collected in a natural setting. Candidate contexts might be tasks requiring typing or tasks for which a wearable (e.g., watch) could be trained to detect on- or off-task activity.

## Author contributions

**Conceptualization:** Kaden Hart, Christopher M. Warren, John Edwards.

**Data curation:** Kaden Hart, John Edwards.

**Formal analysis:** Kaden Hart, Christopher M. Warren, John Edwards.

**Funding acquisition:** John Edwards.

**Investigation:** Kaden Hart, John Edwards.

**Methodology:** Kaden Hart, Christopher M. Warren, John Edwards.

**Project administration:** Kaden Hart, John Edwards.

**Software:** Kaden Hart.

**Supervision:** Kaden Hart, John Edwards.

**Validation:** Kaden Hart.

**Visualization:** Kaden Hart, John Edwards.

**Writing – original draft:** Kaden Hart.

**Writing – review & editing:** Christopher M. Warren, John Edwards.

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
