## [Decision Letter · Decision Letter 0]

26 Mar 2025

PONE-D-24-26297Time-on-task estimation for tasks lasting hours spread over multiple daysPLOS ONE

Dear Dr. Hart,

Thank you for submitting your manuscript to PLOS ONE. After careful consideration, we feel that it has merit but does not fully meet PLOS ONE’s publication criteria as it currently stands. Therefore, we invite you to submit a revised version of the manuscript that addresses the points raised during the review process.

We look forward to receiving your revised manuscript.

Kind regards,

Vineeta Khemchandani

Academic Editor

PLOS ONE

Journal Requirements:

3. Please ensure that you refer to Figures 5-8 in your text as, if accepted, production will need this reference to link the reader to the figure.

Reviewers' comments:

Reviewer's Responses to Questions

**Comments to the Author**

1. Is the manuscript technically sound, and do the data support the conclusions?

Reviewer #1: Yes

Reviewer #2: Yes

2. Has the statistical analysis been performed appropriately and rigorously? 

Reviewer #1: Yes

Reviewer #2: Yes

3. Have the authors made all data underlying the findings in their manuscript fully available?

Reviewer #1: Yes

Reviewer #2: Yes

4. Is the manuscript presented in an intelligible fashion and written in standard English?

Reviewer #1: Yes

Reviewer #2: Yes

5. Review Comments to the Author

Reviewer #1: 1. Might a correction in two data points shown in method section "33 students in an introductory computer programming course (CS1)......", We collected data on 106 student submissions......"

2. The research questions is clearly visible but the objective of the research not written clearly.

3. The implication of the research is to be added.

4. One Correlation Table might be added.

Reviewer #2: Here are few important suggestions for modifications or changes in the manuscript:

1. Enhanced Methodology Section: Expand the methodology section to provide detailed information about the keystroke data collection process, including the specific tools used for recording, the duration of the data collection period, and how participant engagement was defined and measured. This will help readers better understand the rigor of your methods.

2. In-depth Analysis of Results: Include a more thorough analysis of the results, particularly the implications of overestimating time-on-task. Discuss potential psychological or contextual factors that may contribute to this overestimation, providing a deeper understanding of the findings.

3. Integrate Visual Aids: Consider adding visual aids such as graphs or tables that summarize key findings or display trends in time estimation versus actual time spent. This could make the results more accessible and impactful for readers.

4. Broader Implications and Future Directions: In the discussion section, emphasize the broader implications of your findings for educational practices and learning analytics. Suggest specific areas for future research that could expand on your work or explore related questions, thus guiding subsequent studies in the field.

Please site few papers in your manuscript given as follows:

1. Novita, M., Saputro, N. D., Chauhan, A. S., & Waliyansyah, R. R. (2022). Digitalization of Education in the Implementation of Kurikulum Merdeka. KnE Social Sciences, 153-164.

2. Chauhan, A. S. (2022). Modeling and predicting student academic performance in higher education using data mining techniques. International Journal of Software Innovation (IJSI), 10(1), 1-10.

3. Sudirman, S., Rodríguez-Nieto, C. A., Dhlamini, Z. B., Chauhan, A. S., Baltaeva, U., Abubakar, A., ... & Andriani, M. (2023). Ways of thinking 3D geometry: exploratory case study in junior high school students. Polyhedron International Journal in Mathematics Education, 1(1), 15-34.

4. Chauhan, A. S., Singh, Y., & Soam, A. (2012). Effective Decision Making in Higher Educational Institutions Using Data Warehousing and Data Mining. Journal of Computer Science Engineering and Information Technology Research (JCSEITR), 2(1).

6. PLOS authors have the option to publish the peer review history of their article (what does this mean?). If published, this will include your full peer review and any attached files.

Reviewer #1: No

Reviewer #2: No

---

## [Author Response · Author response to Decision Letter 1]

23 Jun 2025

Dear Editors,

We are delighted at the opportunity to revise our paper and even more delighted with the comments we received from reviewers. We believe the paper is a far better offering to the community thanks to their insightful reviews. Below we reproduce specific comments along with the changes we made as a result. The resubmitted paper has a number of changes and additions.

We updated file names and captions. Included .tex file in submition

We note that you have indicated that there are restrictions to data sharing for this study. PLOS only allows data to be available upon request if there are legal or ethical restrictions on sharing data publicly. For more information on unacceptable data access restrictions, please see http://journals.plos.org/plosone/s/data-availability#loc-unacceptable-data-access-restrictions.

Keystroke data is highly identifiable even after removing the identifying labels. This is because students type their name at the beginning of their files, and the keystroke data recreates that name. We can remove this but they have typed their name or other identifying information elsewhere. Even careful inspection can miss potentially identifiable information.

USU’s institutional review board can be contacted at irb@usu.edu or 435-797-1821

Please ensure that you refer to Figures 5-8 in your text as, if accepted, production will need this reference to link the reader to the figure.

Updated references to include in text.

We searched through https://retractiondatabase.org and did not find any retractions.

Reviewer #1: Might a correction in two data points shown in method section "33 students in an introductory computer programming course (CS1)......", We collected data on 106 student submissions......"

Fixed grammar (33 => Thirty-three). Clarified that there are 33 students and 106 assignment submissions (3.2 assignments per student).

Reviewer #1: The research questions is clearly visible but the objective of the research not written clearly.

The objective is to discover how accurate time estimates are for long tasks. To do this we needed to bridge the gap between short term time measures that are objective and long term time measures that are subjective. We added the following sentence to the end of introduction section to clarify this:

“This paper contributes to time-estimate research by demonstrating methodology to obtain objective measures for long duration tasks in a case study exploring subjective time estimates to objective time estimates.”

Reviewer #1: The implication of the research is to be added.

The implication is that better objective measures of time can be found for long duration tasks. This would enable better research into human time perception. See paragraph three of conclusion.

“Our study also enables research into whether additional theories of time awareness apply at larger time scales. For example…”

Reviewer #1: One Correlation Table might be added.

Added Table 2, the correlations we investigated are now easily accessible in one place.

Reviewer #2: Enhanced Methodology Section: Expand the methodology section to provide detailed information about the keystroke data collection process, including the specific tools used for recording, the duration of the data collection period, and how participant engagement was defined and measured. This will help readers better understand the rigor of your methods.

The ShowYourWork plugin was the tool we used for data collection. Added dates for the semesters. Added the engagement formula we used to section “Measurement of Actual Time Duration” and paragraph explaining its use.

Reviewer #2: In-depth Analysis of Results: Include a more thorough analysis of the results, particularly the implications of overestimating time-on-task. Discuss potential psychological or contextual factors that may contribute to this overestimation, providing a deeper understanding of the findings.

Second paragraph of Section Estimated Time is just this. We weren’t testing a specific cause, so I’m including reasons that may contribute.

Added “The perception of doing homework could be worse than the reality of doing homework.” to section Pre-Sruvey

Reviewer #2: Integrate Visual Aids: Consider adding visual aids such as graphs or tables that summarize key findings or display trends in time estimation versus actual time spent. This could make the results more accessible and impactful for readers.

Time estimation vs actual time is fig 6. It is possible that the reviewers couldn’t see the figures because of formatting errors with file names.

Reviewer #2: Broader Implications and Future Directions: In the discussion section, emphasize the broader implications of your findings for educational practices and learning analytics. Suggest specific areas for future research that could expand on your work or explore related questions, thus guiding subsequent studies in the field.

Added the following

1. Students think homework is worse than it is.

2. Investigate other courses in the future, possible one per level: freshmen, sophomore, junior, senior. Do estimation errors change as students become more accustomed to college? Does accuracy change with experience?

3. Future research should investigate the general application of objective time measures for long duration tasks. To do this a model of engagement would need to be made for other contexts. Will self-estimates of task duration be overestimated like homework?

Reviewer #2: Please site few papers in your manuscript given as follows:

1. Novita, M., Saputro, N. D., Chauhan, A. S., & Waliyansyah, R. R. (2022). Digitalization of Education in the Implementation of Kurikulum Merdeka. KnE Social Sciences, 153-164.

2. Chauhan, A. S. (2022). Modeling and predicting student academic performance in higher education using data mining techniques. International Journal of Software Innovation (IJSI), 10(1), 1-10.

3. Sudirman, S., Rodríguez-Nieto, C. A., Dhlamini, Z. B., Chauhan, A. S., Baltaeva, U., Abubakar, A., ... & Andriani, M. (2023). Ways of thinking 3D geometry: exploratory case study in junior high school students. Polyhedron International Journal in Mathematics Education, 1(1), 15-34.

4. Chauhan, A. S., Singh, Y., & Soam, A. (2012). Effective Decision Making in Higher Educational Institutions Using Data Warehousing and Data Mining. Journal of Computer Science Engineering and Information Technology Research (JCSEITR), 2(1).

Response

These papers are not related to time estimation or time estimation errors. We reviewed the first three and did not see the relevance to our research. I was unable to find the fourth. We would be happy to update the citations if we understood how they are related.

---

## [Decision Letter · Decision Letter 1]

6 Aug 2025

Time-on-task estimation for tasks lasting hours spread over multiple days

PONE-D-24-26297R1

Dear Dr. Hart,

We’re pleased to inform you that your manuscript has been judged scientifically suitable for publication and will be formally accepted for publication once it meets all outstanding technical requirements.

Kind regards,

Vineeta Khemchandani

Academic Editor

PLOS ONE

Additional Editor Comments (optional):

Reviewers' comments:

Reviewer's Responses to Questions

**Comments to the Author**

1. If the authors have adequately addressed your comments raised in a previous round of review and you feel that this manuscript is now acceptable for publication, you may indicate that here to bypass the “Comments to the Author” section, enter your conflict of interest statement in the “Confidential to Editor” section, and submit your "Accept" recommendation.

Reviewer #1: All comments have been addressed

2. Is the manuscript technically sound, and do the data support the conclusions?

Reviewer #1: Yes

3. Has the statistical analysis been performed appropriately and rigorously? 

Reviewer #1: Yes

4. Have the authors made all data underlying the findings in their manuscript fully available?

Reviewer #1: Yes

5. Is the manuscript presented in an intelligible fashion and written in standard English?

Reviewer #1: Yes

6. Review Comments to the Author

Reviewer #1: No more comments from my site. The authors addressed all the comments. Accept the manuscript. This manuscript has highlighted some clear insight.

7. PLOS authors have the option to publish the peer review history of their article (what does this mean?). If published, this will include your full peer review and any attached files.

Reviewer #1: **Yes: **Aurobindo Kar

---

## [Editor Report · Acceptance letter]

PONE-D-24-26297R1

PLOS ONE

Dear Dr. Hart,

I'm pleased to inform you that your manuscript has been deemed suitable for publication in PLOS ONE. Congratulations! Your manuscript is now being handed over to our production team.

Kind regards,

on behalf of

Dr. Vineeta Khemchandani

Academic Editor

PLOS ONE